


# Observational constraints on methane emissions from Polish coal mines using a ground-based remote sensing network

Andreas Luther[1], Julian Kostinek[8], Ralph Kleinschek[1], Sara Defratyka[6], Mila Stanisavljević[4], Andreas Forstmaier[3], Alexandru Dandocsi[5], Leon Scheidweiler[1], Darko Dubravica[2], Norman Wildmann[8], Frank Hase[2], Matthias M. Frey[7], Jia Chen[3], Florian Dietrich[3], Jarosław Nęcki[4], Justyna Swolkień[4], Christoph Knote [9], Sanam N. Vardag[1,10], Anke Roiger[8], and André Butz[1,10,11]

[1]Institute of Environmental Physics (IUP), Heidelberg University, Heidelberg, Germany
[2]Karlsruhe Institute of Technology (KIT), Institute of Meteorology and Climate Research (IMK-ASF), Karlsruhe, Germany
[3]Environmental Sensing and Modeling, Technical University of Munich (TUM), Munich, Germany
[4]AGH - University of Science and Technology, Kraków, Poland
[5]National Institute of Research and Development for Optoelectronics (INOE2000), Măgurele, Romania
[6]Laboratoire des sciences du climat et de l'environnement (LSCE-IPSL) CEA-CNRS-UVSQ Université Paris Saclay, Gif-sur-Yvette, France
[7]National Institute for Environmental Studies, Tsukuba, Japan
[8]Deutsches Zentrum für Luft- und Raumfahrt (DLR), Institut für Physik der Atmosphäre, Oberpfaffenhofen, Germany
[9]Model-Based Environmental Exposure Science, University of Augsburg, Augsburg, Germany
[10]Heidelberg Center for the Environment (HCE), Heidelberg University, Heidelberg, Germany
[11]Interdisciplinary Center for Scientific Computing (IWR), Heidelberg University, Heidelberg, Germany

**Correspondence:** Andreas Luther (andreas.luther1@gmail.com), André Butz (andre.butz@iup.uni-heidelberg.de)

**Abstract.** Given its abundant coal mining activities, the Upper Silesian Coal Basin (USCB) in southern Poland is one of the largest sources for anthropogenic methane ($CH_4$) emissions in Europe. Here, we report on $CH_4$ emission estimates for coal mine ventilation facilities in the USCB. Our estimates are driven by pair-wise upwind-downwind observations of the column-average dry-air mole fractions of $CH_4$ ($XCH_4$) by a network of four portable, ground-based, sun-viewing Fourier Transform Spectrometers of the type EM27/SUN operated during the CoMet campaign in May/June 2018. The EM27/SUN were deployed in the four cardinal directions around the USCB in approx. $50\,\mathrm{km}$ distance to the center of the basin. We report on six case studies for which we inferred emissions by evaluating the mismatch between the observed downwind enhancements and simulations based on trajectory calculations releasing particles out of the ventilation shafts using the Lagrangian particle dispersion model FLEXPART. The latter was driven by wind fields calculated by WRF (Weather Research and Forecasting model) under assimilation of vertical wind profile measurements of three co-deployed wind lidars. For emission estimation, we use a Phillips-Tikhonov regularization scheme with the L-curve criterion. Diagnosed by the averaging kernels, we find that, depending on the catchment area of the downwind measurements, our ad-hoc network can resolve individual facilities or groups of ventilation facilities but that inspecting the averaging kernels is essential to detected correlated estimates. Generally, our instantaneous emission estimates range between 80 and $133\,\mathrm{kt}$ $CH_4$ $\mathrm{a}^{-1}$ for the south-eastern part of the USCB and between 414 and $790\,\mathrm{kt}$ $CH_4$ $\mathrm{a}^{-1}$ for various larger parts of the basin, suggesting higher emissions than expected from the

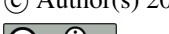



annual emissions reported by the E-PRTR (European Pollutant Release and Transfer Register). Uncertainties range between 23 and 36% dominated by the error contribution from uncertain wind fields.

# 1 Introduction

The atmospheric abundance of methane ($CH_4$) increased by a factor of 2.6 since pre-industrial times from roughly $720\,ppb$
(parts-per-billion) to about $1879\,ppb$ in 2020 (Dlugokencky, 2021) mainly driven by anthropogenic influences (e.g. Bousquet et al., 2006; Loulergue et al., 2008; Kirschke et al., 2013; IPCC, 2013; Nisbet et al., 2014; Conley et al., 2016; Schwietzke et al., 2016; Worden et al., 2017; Alvarez et al., 2018; Saunois et al., 2020; Hmiel et al., 2020). Roughly $20\,\%$ of the total, global anthropogenic $CH_4$ emissions are caused by the fossil fuel industry (Bousquet et al., 2006; Schwietzke et al., 2016; Saunois et al., 2020) and an extensive source of $CH_4$ is hard coal mining. Poland is the largest hard coal producer in the European
Union with the Upper Silesian Coal Basin (USCB) as one of the largest hard coal producing regions in Europe. Several bottom-up inventories report on the total $CH_4$ emissions for the USCB: According to the GESAPU database, the USCB emitted a total of $405\,kt\ CH_4$ in 2010 (Bun et al., 2019). The E-PRTR (European Pollutant Release and Transfer Register, http://prtr.ec.europa.eu/, 2018) collects emission reports of every individual mine yielding an aggregated total of $507\,kt\ CH_4$ $a^{-1}$ for the USCB. Dreger (2021) reports hard coal mining emissions of $530\,kt\ CH_4$ for the USCB in 2018 and the Copernicus
Atmosphere Monitoring Service regional emission inventory (CAMS-REG-GHG/AP) lists $632\,kt\ CH_4\ a^{-1}$ (Granier et al., 2019; Fiehn et al., 2020). EDGAR v4.3.2 (Emission Database for Global Atmospheric Research) accounts emissions of $720\,kt$ $CH_4\ a^{-1}$ (Janssens-Maenhout et al., 2017) for the USCB in the year 2017.

In addition to the bottom-up inventories, top-down approaches examined the USCB emissions. During the CoMet mission (Carbon dioxide and Methane mission 2018, from 23 May to 12 June 2018), several ground-based instruments and aircraft
measured the atmospheric $CH_4$ abundance in the USCB. In our precursor study (Luther et al., 2019), we used stop-and-go measurements of the column-average dry-air mole fractions of $CH_4$ ($XCH_4$) by a mobile, ground-based Fourier Transform Spectrometer (FTS) to evaluate the mining emissions of individual ventilation facilities, and found similar emissions as suggested by the E-PRTR inventory. The total USCB emission estimates of Fiehn et al. (2020) and Kostinek et al. (2020), based on airborne in situ measurements, are in broad agreement with the E-PRTR data for single flights. Using airborne imager data,
Krautwurst et al. (2021) found some discrepancies between their estimates and the E-PRTR inventory for small groups of ventilation facilities. The isotopic $CH_4$ composition was measured by Menoud et al. (2021) with ground-based in situ instruments. Swolkień (2020) discusses the short-term, shaft-wise $CH_4$ release in the USCB.

Given the magnitude of emissions and the range of estimates, $CH_4$ in the USCB warrants further investigation. Here, we report on $CH_4$ emission estimates derived from measurements of four stationary, sun-viewing FTS of the type EM27/SUN
arranged in a network-like pattern enclosing the USCB during the CoMet campaign activities. The setup largely mimics previous network deployments for quantifying urban greenhouse gas emissions in Berlin (Hase et al., 2015), Paris (Vogel et al., 2019), St. Petersburg (Makarova et al., 2020), Munich (Dietrich et al., 2021), Indianapolis (Jones et al., 2021) and other places. Our four EM27/SUN were positioned in the four cardinal directions at a distance of a few tens of kilometers to the



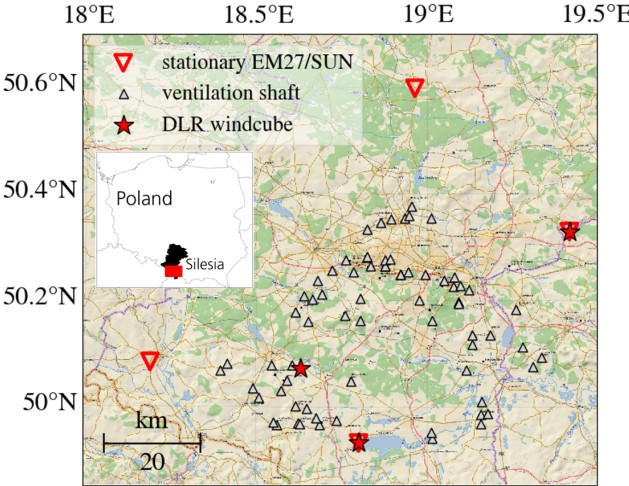

**Figure 1.** The USCB in southwest Poland. The small inset on the left illustrates the region of Silesia in black and the map excerpt of the USCB in red). Ventilation shafts are marked as gray triangles. Stationary EM27/SUN FTS locations are marked as red triangles; the three wind lidars DLR85, DLR86, and DLR89 are marked as red stars. Eastern and southern wind lidars are placed at the same locations as the respective EM27/SUNs. The western wind lidar is located about 30 km to the east of the western EM27/SUN. Background map from ESRI (2019).

center of the USCB. We calculate differences between pairs of upwind and downwind observations to determine enhancements
($\Delta$XCH$_4$) in our XCH$_4$ records attributable to sources within the USCB. Developing on the model setup of Kostinek et al.
(2020), we use the Flexible Lagrangian particle dispersion model (FLEXPART) together with wind fields from the Weather
Research and Forecasting model (WRF) constrained by three wind lidars to translate the observed XCH$_4$ enhancements into
emission estimates for groups of coal mine ventilation shafts. To this end, we set up an inverse estimation scheme based on
Phillips-Tikhonov regularization. This setup enables careful information content analysis while not relying on estimates of a
priori emission uncertainties which are often inaccessible.

Our manuscript first summarizes the campaign setup (Sect. 2). Then, Sect. 3 and 4 detail the modeling and regularization
methods. Sect. 5 reports on emission estimates for six case studies and Sect. 6 discusses our results in terms of compatibility
with other emission estimates and methodological strengths and weaknesses.

## 2  Campaign deployment and XCH$_4$ measurements

As part of the CoMet activities in May and June 2018, our campaign deployment of the EM27/SUN focused on the USCB
extending roughly $80 \times 80$ km$^2$ in the southwest of Poland. Fig. 1, adopted from our precursor study on mobile measurements
(Luther et al., 2019), illustrates the network pattern together with the locations of the most important coal mine ventilation
shafts and the wind lidars. The four stationary EM27/SUN spectrometers are positioned roughly in the four cardinal directions
around the center of the basin, ensuring that at least one instrument measures upwind and one downwind of the USCB for most





wind situations. Due to mainly easterly wind conditions prevailing during our deployment, the eastern station (The Glade) functions as the upwind station for all case studies discussed here. The respective downwind stations are located in the south (Pustelnik) and west (Raciborz) of the USCB. The northern station (Za Miastem) was identified neither upwind nor downwind station for any of the cases.

The functioning of the EM27/SUN FTS and the data reduction techniques are described in detail by Gisi et al. (2012), Hase
et al. (2015) and Hase et al. (2016) and with a particular focus on our setup by Luther et al. (2019). Generally, the EM27/SUN observe spectra of direct sunlight in the shortwave infrared spectral range from which the total column concentrations of $CH_4$, carbon dioxide ($CO_2$), water vapor ($H_2O$), molecular oxygen ($O_2$), carbon monoxide (CO), and other substances (Butz et al., 2017) can be retrieved. $CH_4$ and $O_2$ are of relevance here. All EM27/SUN spectrometers participating in the campaign successfully underwent the instrumental quality assurance tests required by the Collaborative Carbon Column Observing Network
(COCCON) and presented in Frey et al. (2019) before field deployment.

We run the software package PROFFIT (Hase et al., 2004) to retrieve the column concentrations $[O_2]$ and $[CH_4]$ from the $7765$ to $8005\,cm^{-1}$ and $5897$ to $6145\,cm^{-1}$ spectral windows, respectively, using absorption line parameters by Toon (2017) and Rothman et al. (2009). The respective column-averaged dry-air mole fractions of methane, $XCH_4$, are calculated via normalization through $\frac{[CH_4]}{[O_2]} \times 0.2094$ where the atmospheric $O_2$ mole fraction is assumed $0.2094$. The normalization is generally
recommended to lessen spurious artefacts due to pressure and solar zenith angle (SZA) dependencies. The SZAs during our measurements did not exceed $56°$ which is within the range which does not require airmass-dependent bias correction (Frey et al., 2015). Slightly deviating from the processing recipe (Frey et al., 2019) and in line with our precursor study (Luther et al., 2019), our $CH_4$ retrievals only scale the $CH_4$ concentrations in the layers below $1700\,m$ a.g.l. where we expect elevated methane due to the localized sources at the ground. Further, we extract the a priori $CH_4$ profiles from a dedicated
ECHAM/MESSy Atmospheric Chemistry simulation described by Jöckel et al. (2016) and Nickl et al. (2020).

For network deployments such as undertaken here, it is common practice to cross-calibrate the network nodes by side-by-side measurements (Frey et al., 2015; Chen et al., 2016; Frey et al., 2019; Jones et al., 2021; Dietrich et al., 2021) in order to exclude spurious gradients when conducting upwind-downwind analyses. We calibrated the four instruments through side-by-side measurements on 23 May and 26 May, 2018 at the southern location Pustelnik. Fig. 2 shows the raw and calibrated $XCH_4$
records and Table 1 lists the respective calibration factors. These instrument-specific empirical calibration factors have been applied to the $XCH_4$ campaign data discussed in the following. After calibration, the mean, absolute instrument-to-instrument difference is roughly $1\,ppb$ which is compatible to Frey et al. (2019) who state the instrument-to-instrument difference for an EM27/SUN ensemble with $0.8\,ppb$ for $XCH_4$. We estimate the precision for $XCH_4$ with $0.6\,ppb$ for one minute integration time from measurements sampled during the campaign in the rather variable methane field of the USCB, which is consistent
with the precision stated by Chen et al. (2016). Based on measurements of the eastern and northern instruments (The Glade and Za Miastem) observed from $7\,UTC$ to $10\,UTC$ on 28 May 2018 (upper panel Fig. 7), we calculated the precision as the standard deviation of the observations, averaged between the two instruments during this period. We chose this period, since the two timelines are not affected by any strong methane sources in the vicinity.



In addition to the EM27/SUN network, we also operated three Leosphere Windcube 200S Doppler wind lidars (Vasiljević
et al., 2016; Wildmann et al., 2018, 2020) marked as red stars in Fig. 1 and as detailed in Luther et al. (2019) and Kostinek et al.
(2020). The measured wind profiles (10 min time interval) reach up to 4 km a.g.l. and are assimilated into the WRF simulations
to improve the modeled wind fields as discussed in Sect. 3.

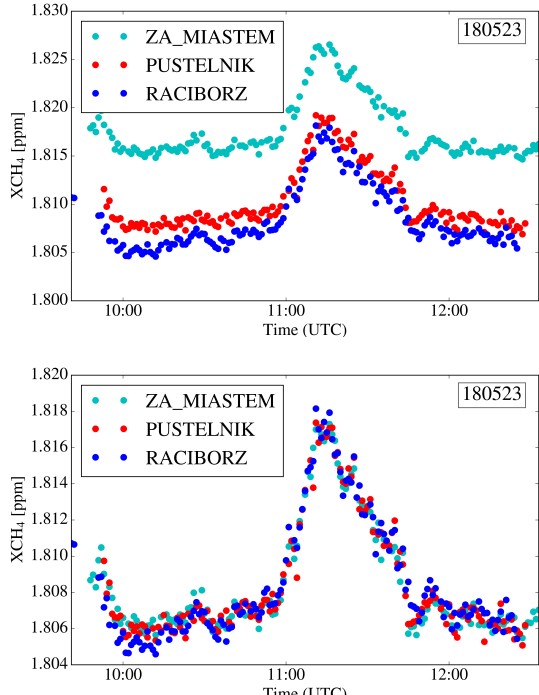

**Figure 2.** Side-by-side measurements at station Pustelnik on 23 May 2018 on the first day of the deployment. The upper panel displays
unscaled data with an average instrument-to-instrument difference of 5 ppb. The lower panel shows measured XCH$_4$ after scaling with about
1 ppb instrument-to-instrument difference. Note, that the instruments detected a plume like structure at the beginning of the measurements
and around 11:20 UTC. The instrument The Glade measured side-by-side on 26 May with the Pustelnik instrument (data in A1).

| Site | Lat. °N | Lon. °E | m a.s.l | XCH$_4$ cal. | O$_2$ cal. |
|------|---------|---------|---------|--------------|------------|
| Za Miastem (N) | 50.599 | 18.963 | 305 | 0.9949 | 1.0079 |
| The Glade (E) | 50.329 | 19.416 | 303 | 1.0002 | 0.9967 |
| Pustelnik (S) | 49.933 | 18.799 | 266 | 0.9989 | 1.0034 |
| Raciborz (W) | 50.083 | 18.192 | 223 | 1 | 1 |

**Table 1.** Calibration factors towards the western station Raciborz for each instrument and species, nominal geolocation in network.



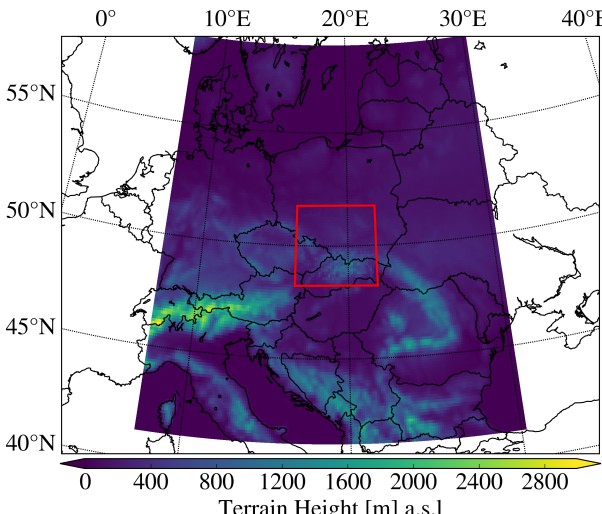

**Figure 3.** Overview of the two WRF domains over central Europe. The first, larger domain (colored shape) has a spatial resolution of $\sim 15\,\text{km}$. The inner domain (red rectangle) has a spatial resolution of $\sim 3\,\text{km}$ and is focused on the USCB. The Tatra mountain ridge (with elevations $> 2000\,\text{m a.s.l}$) in the southern part of the second domain along the border towards Slovakia is fully enclosed by the inner domain.

## 3  Dispersion modeling of methane

Our simulations of methane dispersion in the USCB partition into two steps largely adopting the basic setup reported by
Kostinek et al. (2020): first (Sect. 3.1), the wind fields are modeled by a two domain WRF setup including assimilation of the wind lidar observations. Second, the WRF wind fields drive the particle dispersion in the Lagrangian particle dispersion model FLEXPART (Sect. 3.2).

### 3.1   WRF wind fields

The WRF V4 (Skamarock et al., 2019) setup is driven by 3-hourly GFS data (NCEP, 2017) in two domains (Fig. 3) focusing
on central Europe and the USCB, respectively. The outer domain has a spatial resolution of roughly $15\,\text{km}$, the inner domain has a spatial resolution of roughly $3\,\text{km}$. The simulations start at $00\,\text{UTC}$ on the day of interest to allow for several hours of spin-up. Details are explained in Kostinek et al. (2020).

WRF has the possibility to assimilate observational data via Four Dimensional Data Assimilation (FDDA), which we use for our wind lidar measurements. On a ten minute time interval, the wind profile observations are fed into the WRF calculations
to constrain the simulated wind fields. The assimilation process is adjustable via several parameters, e.g. radius of influence $r_{xy}$, time window $\Delta t$ and horizontal wind coefficient $c_{uv}$ in the WRF input file directly. Kostinek et al. (2020) have identified a selection of parameter settings to obtain the best-guess parameter combinations for the same region, WRF domains and time periods discussed here. Therefore, we adopt the same setup here and report on the overall WRF performance by comparing the simulations with the wind lidar observations.



Fig. 4 displays a comparison between modeled and observed wind speed (upper panel) and wind direction (lower panel) for altitude levels in the planetary boundary layer (PBL). PBL height is estimated by means of eddy dissipation rate gradients calculated from the wind lidar observations directly. The observation levels of the wind lidar do not match the WRF grid levels exactly. Thus, the comparison includes modeled and observed data if the WRF wind level is within the range of $\pm 25\,\mathrm{m}$ around the level of observation. This represents all of the WRF wind levels but dismisses some of the wind lidar levels as the lidar

data has a finer vertical resolution. The comparison is restricted to the three days of the case studies to be discussed below (28 May; 6 June; 7 June, 2018). The prevailing wind directions were northeast to southeast and prevailing wind speeds range from $2\,\mathrm{m\,s^{-1}}$ to $9\,\mathrm{m\,s^{-1}}$. The root-mean-square-error (RMSE) for the wind speed comparison calculates to $1.7\,\mathrm{m\,s^{-1}}$. When eliminating the top two levels, the wind direction RMSE reduces from $38°$ to $26°$ indicating model inaccuracies towards the PBL top where wind shear effects are expected. Low wind speeds on 7 June are possibly deteriorating the wind direction bias,

as wind direction uncertainties are generally larger for low wind speeds. Further challenges for the wind direction estimation are the onset of convection during the observational morning hours with subsequent PBL rise and the calming winds towards the end of the observational days. In general we observed most outliers for the top comparison levels for wind direction, which could be related to a significant number of conspicuous low wind speed simulations and observations for these levels. This might be related to model uncertainties when estimating the PBL height, leading to misinterpretation of actual above-PBL

observations with inside-PBL simulations and vice versa.

### 3.2   Lagrangian methane dispersion via FLEXPART

WRF windfields drive the trajectory calculations in FLEXPART. The model simulates trajectories for $50\,000$ particles for every USCB coal mining shaft reported by the E-PRTR and the CoMet database (Gałkowski et al., 2021) with a total mass of $10^5\,\mathrm{kg}$ $CH_4$. The simulations do not consider background $CH_4$. The model releases particles in a $10\,\mathrm{m} \times 10\,\mathrm{m} \times 10\,\mathrm{m}$ box

on the ground. The modeling period starts at 00:10 UTC the day of interest and continues until 17:50 UTC which results in 17.7 h simulation time. We chose the grid output option in FLEXPART with $100 \times 100$ boxes and a spatial resolution of roughly 1.3 km stacked in 24 layers up to 3 km altitude. The simulated $XCH_4$ measurements are the sums of all boxes above each pixel enclosing an EM27/SUN location. The 6 min FLEXPART output is interpolated to the observational time interval which generally is one measurement per minute. After unit conversion, the simulated methane enhancement is compared to the

measured upwind-downwind difference $\Delta XCH_4$.

    The FLEXPART simulations are iterated with slightly different meteorological parameters to provide an uncertainty analysis. There are seven ensemble runs: the `CONTROL` run with best guess input, the `WINDp5` run with $+5°$ wind direction change of the whole wind field, the `WINDm5` run with $-5°$ wind direction change of the whole wind field, the `SPEEDp06` run with the wind speed increased by $0.9\,\mathrm{m\,s^{-1}}$, the `SPEEDm06` run with the wind speed decreased by $0.9\,\mathrm{m\,s^{-1}}$, the `PBLp100` run with

the PBL height increased by 100 m, and the `PBLm100` run with the PBL height decreased by 100 m. We use the same ensemble set up as discussed by Kostinek et al. (2020).

    We further used the FLEXPART simulations for modeling the air mass travel time from the upwind to the downwind instruments (Fig. 5). To this end, we implemented a virtual methane source at the upwind measurement location at the beginning

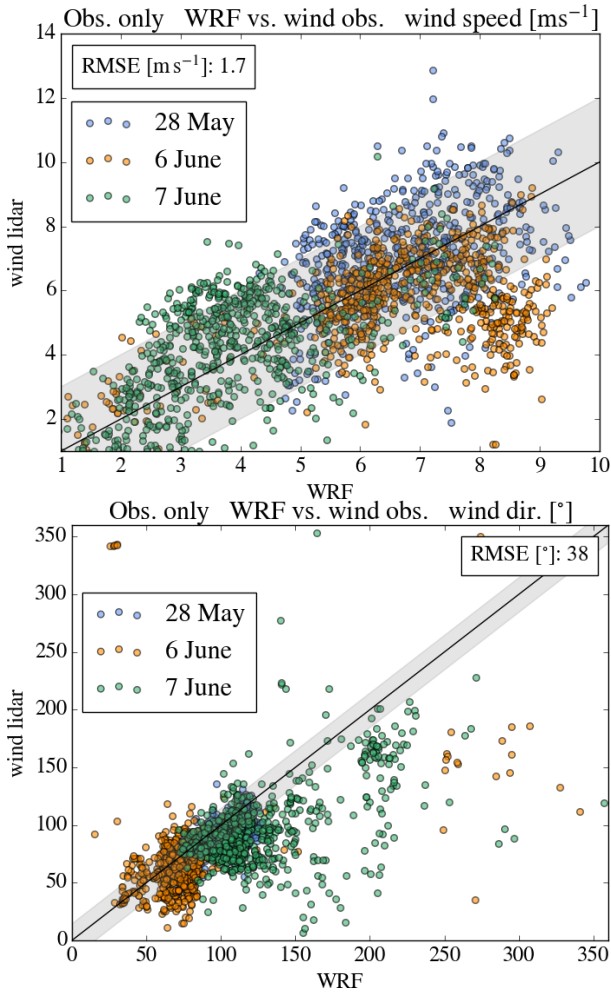

**Figure 4.** Comparison of WRF wind estimates and wind lidar observations for the observational period between 7 UTC and 17 UTC for the three days of interest 28 May (blue), 6 June (orange), and 7 June (green). Shaded areas include roughly 80 % of the data points for $\pm 2\,\mathrm{m\,s^{-1}}$ for wind speed in the upper panel and include roughly 50 % of wind direction measurements in a $\pm 15°$ range in the lower panel. Wind direction simulations differ most from the observations for 7 June (green), a day with generally lower wind speeds, than the other two discussed cases. This indicates wind information uncertainties, when it comes to low wind speeds. For the wind direction comparison, note, that values close to $0°$ and $360°$ represent virtually the same wind direction but may introduce an error to the root mean square error (RMSE) calculations.

of each observational period. Then, we estimated the airmass travel time by recording the time required for the virtual tracer

plume to cross the tangent point of the downwind measurement location i.e. the point that was closest to the downwind location along the PBL averaged wind direction. Given that the distance between upwind and downwind locations exceeds 50 km for some cases, estimated travel times range between 1.5 h and 4.6 h. Thus, in order for the upwind measurement to be





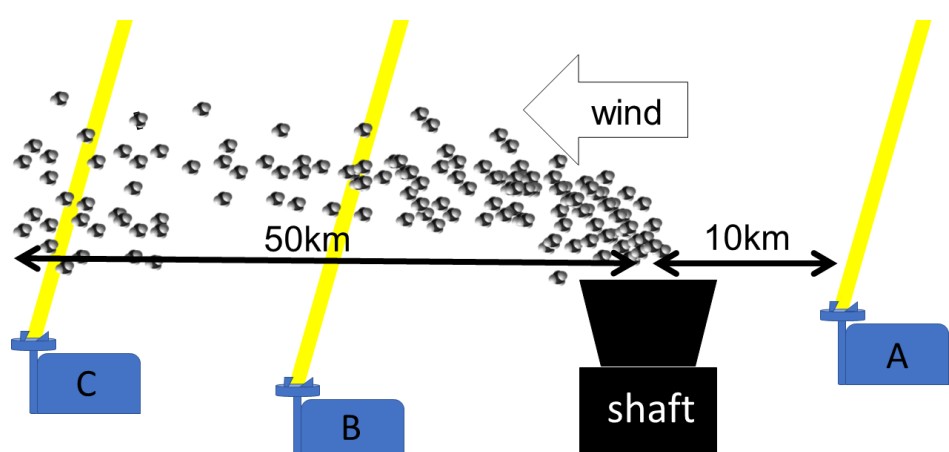

**Figure 5.** Lagrangian time lag sketch. Instrument A measures background methane (upwind). The methane enhancements induced by the emissions of the shaft are calculated by subtracting the background (upwind) measurements from the downwind observations of instruments B and C. Depending on wind speed, the air mass measured by instrument A will be at instrument B and C after different travel times.

representative of the background conditions for the later downwind measurement, we consider the airmass travel time between the two measurements by subtracting the respective time-shifted upwind measurements when calculating $\Delta$XCH$_4$. The travel

time is calculated once for every case study beginning with the first upwind measurement which is part of the inversion process and ending when the virtual tracer reaches the respective downwind instrument. To account for possible background methane variability, we altered the simulated travel time by $\pm 30$ min and calculated the average methane enhancement for each individually time-shifted period. The difference between minimum and maximum averaged $\Delta$XCH$_4$ of these periods is included in the emission estimation routine to represent the error related to background methane uncertainties.

**4    Emission estimates via Phillips-Tikhonov regularization**

In order to estimate the shaft emissions from the mismatch between measured and simulated methane enhancements $\Delta$XCH$_4$, we use a Phillips-Tikhonov inverse method. We set it up such that the state vector $\boldsymbol{x}$ consists of $m$ dimensionless factors that scale the emissions of each coal mine ventilation shaft considered by the FLEXPART simulations. We assume the scaling factors to be constant for each day of measurement i.e. we impose the assumption of the source strength being constant over the

time of a day. FLEXPART is the forward model $\boldsymbol{K}$ ($m \times n$) relating the emissions of the $n$ shafts to $m$ $\Delta$XCH$_4$ measurements. The measurement vector $\boldsymbol{y}$ contains the $\Delta$XCH$_4$ enhancements, observed at 1 min intervals, translated from units ppb into total mass column enhancements kg m$^{-2}$. The Phillips-Tikhonov inverse method then delivers the estimated state vector $\boldsymbol{x_\lambda}$ by minimizing a two-term cost function consisting of a measurement term and an a priori term (Phillips, 1962; Tikhonov, 1963; Twomey, 1963):

$$\boldsymbol{x_\lambda} = argmin \left\{ \|\boldsymbol{S}_\epsilon^{-\frac{1}{2}}(\boldsymbol{K}\boldsymbol{x} - \boldsymbol{y})\|_2^2 + \lambda^2 \|\boldsymbol{W}(\boldsymbol{x} - \boldsymbol{x}_a)\|_2^2 \right\} \quad\quad (1)$$



with $\boldsymbol{S}_\epsilon$ the error covariance matrix, $\lambda$ the regularization parameter, $\boldsymbol{W}$ the weighting operator, $\boldsymbol{x}_a$ the a priori state vector, and $||\cdot||_2$ representing the $L_2$ norm. $\boldsymbol{S}_\epsilon$ contains the averaged standard deviation of the FLEXPART simulation ensemble summed in quadrature with the XCH$_4$ background variability and the measurement noise. The latter is assumed to amount to $0.6\,\mathrm{ppb}$ which corresponds to the standard deviation of the averaged background measurements of two stationary instruments (The Glade and Za Miastem) from $7\,\mathrm{UTC}$ to $10\,\mathrm{UTC}$ on 28 May 2018 (see top panel in Fig. 7). The estimated background variability ranges between $0.3\,\mathrm{ppb}$ and $2.2\,\mathrm{ppb}$ for the individual case studies, based on the differences of minimum and maximum average $\Delta$XCH$_4$ calculated under consideration of $\pm 30\,\mathrm{min}$ time shifts of the simulated methane travel times. $\boldsymbol{W}$ is a diagonal matrix with elements $\frac{1}{x_{a,j}}, j=1,\ldots,m$ which renders the second cost term dimensionless (Butz et al., 2012). Technically, we transform the Phillips-Tikhonov regularization problem into a plain least-squares fit using the definitions (Hansen and O'Leary, 1993; Hansen, 1999; Golub and Von Matt, 1997),

$$\boldsymbol{C} = \begin{bmatrix} \boldsymbol{S}_\epsilon^{-\frac{1}{2}}\boldsymbol{K} \\ \lambda\boldsymbol{W} \end{bmatrix} \text{ and } \boldsymbol{d} = \begin{bmatrix} \boldsymbol{S}_\epsilon^{-\frac{1}{2}}\boldsymbol{y} \\ \lambda\boldsymbol{W}\boldsymbol{x}_a \end{bmatrix} \tag{2}$$

which transforms equ. (1) to

$$\boldsymbol{x}_\lambda = argmin\left\{ \|\boldsymbol{C}\boldsymbol{x}-\boldsymbol{d}\|_2^2 \right\} \tag{3}$$

treatable by a standard least-squares solver (e.g. `python` module `scipy.optimize.lsq_linear`).

The a priori information $\boldsymbol{x}_a$ for each ventilation shaft is taken from the annual E-PRTR emission inventory updated by Gałkowski et al. (2021). The a priori generally guides the minimization process towards physically reasonable solutions if the inverse problem tends to be ill-posed e.g. when there is insufficient measurement information on some of the state vector elements. However, the solution is dependent on the regularization parameter $\lambda$ which has to be found by trading the propagation of measurement errors against influence of the a priori. Here, we use the L-curve criterion to determine the regularization parameter $\lambda$ for each individual case study (e.g. Hansen, 1999). The L-curve is a graphical representation of the the mismatch between measurements and simulations $\|\boldsymbol{K}\boldsymbol{x}_\lambda - \boldsymbol{y}\|_2$ plotted against the norm of the state vector $\|\boldsymbol{x}_\lambda\|_2$ evaluated for a range of $\lambda$ (cf. Fig. 9). The plot typically looks like an L. For small $\lambda$, the measurement term dominates the cost function, the estimate $\boldsymbol{x}_\lambda$ becomes noisy and drives large deviations from the a priori. For large $\lambda$, the a priori term dominates the cost function, the estimate $\boldsymbol{x}_\lambda$ ignores the measurements and produces a large norm of the measurement term. The corner of the L indicates a reasonable regularization parameter for the given minimization problem and is graphically chosen. In addition, we found that the shape of the L-curve is sensitive to forward model errors e.g. when errors in the FLEXPART trajectories and the driving wind fields suggest a spurious, erroneous link between emissions and methane enhancements. Obvious distortions of the L-curve shape are used as a criterion to filter out periods, when the forward model does not represent the actual dispersion conditions.

Besides the L-curve, the regularized inversion approach holds another diagnostic measure: the averaging kernel matrix $\boldsymbol{A}_\lambda$ with dimensions $m \times m$. It is defined via the gain matrix $\boldsymbol{G}_\lambda$ (Rodgers, 2000; Butz et al., 2012; Borsdorff et al., 2014),

$$\boldsymbol{A}_\lambda = \boldsymbol{G}_\lambda\boldsymbol{K} \tag{4}$$

$$\boldsymbol{G}_\lambda = \left(\boldsymbol{K}^T\boldsymbol{S}_\epsilon^{-1}\boldsymbol{K} + \lambda^2 \cdot \boldsymbol{W}^T\boldsymbol{W}\right)^{-1}\boldsymbol{K}^T\boldsymbol{S}_\epsilon^{-1} \tag{5}$$





The averaging kernel matrix, for a given regularization strength $\lambda$, diagnoses how information propagates from the true and a priori states, $\boldsymbol{x}_{true}$ and $\boldsymbol{x}_a$, into the emission estimate,

$$\boldsymbol{x}_\lambda = \boldsymbol{A}_\lambda \boldsymbol{x}_{true} + (\boldsymbol{W} - \boldsymbol{A}_\lambda) \boldsymbol{x}_a \tag{6}$$

The rows of $\boldsymbol{A}_\lambda$ are called the averaging kernels quantifying how an estimated state vector element calculates from the other state elements and what portion comes from the prior. For our purposes, the averaging kernels quantify how the emission estimate for a ventilation shaft is affected by the neighboring shafts and whether there is sufficient measurement information. In the perfect case, the averaging kernel is unity for the shaft under consideration and zero for all other shafts indicating that

the shaft can be perfectly resolved and discriminated from neighboring sources and that it is well-constrained by measurement information. In reality, groups of neighboring sources and sources behind each other along the trajectory are not resolvable and some shafts only marginally affect our measurements implying broader and smaller averaging kernels.

The errors due to measurement noise, background methane variability, and atmospheric transport incorporated in $\boldsymbol{S}_\epsilon$ are propagated into the a posteriori error covariance for the emission estimates via

$$\boldsymbol{S}_{x,\lambda} = \boldsymbol{G}_\lambda \boldsymbol{S}_\epsilon \boldsymbol{G}_\lambda^T \tag{7}$$

where we report the square-root of the diagonal as the error bars of the shaft-wise emission estimates and the square-root of the sum of the entire covariance matrix as the error of the total emissions aggregated over all shafts. For the case studies discussed in Sect. 5, the emission errors due to measurement noise range between $0.62\,\mathrm{kt\,a^{-1}}$ and $4.46\,\mathrm{kt\,a^{-1}}$ which is small compared to the errors introduced by the dispersion modeling ensemble which range between $27\,\mathrm{kt\,a^{-1}}$ and $143\,\mathrm{kt\,a^{-1}}$.

Errors related to background methane variability introduced by a $\pm 30\,\mathrm{min}$ time shift of the airmass travel time range between $0.83\,\mathrm{kt\,a^{-1}}$ and $8.5\,\mathrm{kt\,a^{-1}}$.

## 5 Case studies

We report on six case studies on three different days of the CoMet campaign. For the respective three days 28 May, 6 June and 7 June, 2018, Fig. 6 illustrates typical FLEXPART trajectories of air masses around midday dispersing out of the USCB coal

mine ventilation shafts. For all cases, easterly winds led to the southern station Pustelnik being influenced by a few southern shafts (red trajectories) and the western station Raciborz being influenced by many shafts in various parts of the basin (blue trajectories). The eastern station The Glade provides the background measurements, the northern station Za Miastem was not used here since The Glade was the better background station given the prevailing easterly winds.

Fig. 7 depicts the corresponding $XCH_4$ measurements for all stations, indeed, pointing at significantly elevated concentra-

tions at the downwind sites typically amounting to $\Delta XCH_4$ on the order of $10\,\mathrm{ppb}$ with some diurnal and day-to-day variability. For 28 May, the maxima during the morning hours at Pustelnik and Raciborz are most likely connected to night-time methane accumulation and subsequent transport with rising convection and mixing during the morning hours. When considering the



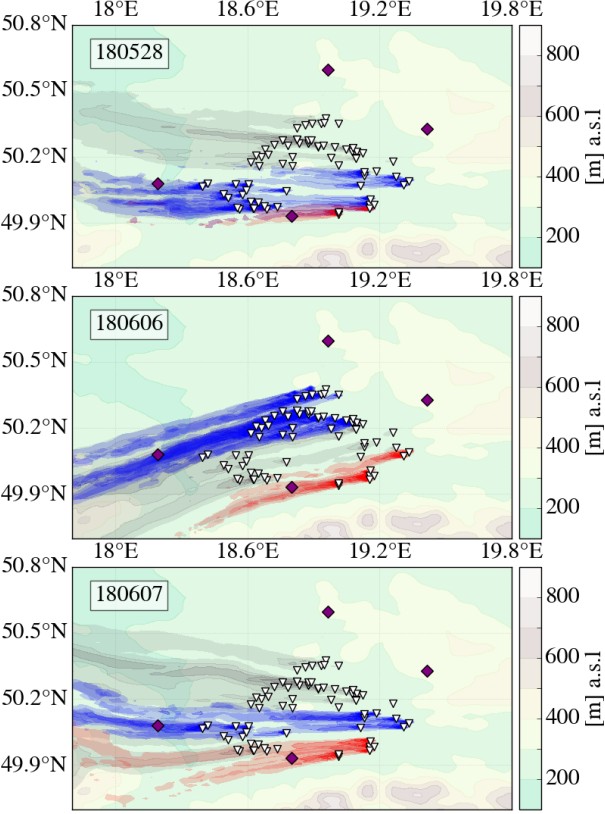

**Figure 6.** FLEXPART simulations of $CH_4$ trajectories released from the USCB ventilation shafts for 28 May (top), 6 June (center), and 7 June, 2018 (bottom, simulation at 12:04 UT). Red, blue and grey shadings indicate trajectories that went over Pustelnik, Raciborz or none of the observation sites, respectively. Background shading indicates topography.

Lagrangian travel time of air masses, these morning hours are excluded from the period of investigation since we have to wait for the air masses from the upwind site The Glade to arrive downwind at Pustelnik and Raciborz. Therefore, the period of in-
vestigation starts later than the Pustelnik and Raciborz measurement records. For Pustelnik, travel times ranged between $1.3\,h$ and $2\,h$, and for Raciborz between $3\,h$ and $5.5\,h$, where 7 June had the longest travel times linked to the slow wind speeds on that day (cf. Fig. 4). In particular, for 7 June, the consideration of the travel time implies that only a small fraction of the measurements at the end of the day is considered for the emission estimates. The stations The Glade and Za Miastem show roughly consistent background $XCH_4$ for 28 May and 6 June, but, on 7 June, significant differences indicate that the background con-
centrations field was not homogeneous. Nonetheless, the FLEXPART trajectories point to The Glade being representative of the background conditions for the downwind stations Raciborz and Pustelnik. Following the procedures described in Sect. 3.2 and 4, we calculate observed $\Delta XCH_4$ from the $XCH_4$ observations by subtracting the background record of The Glade from the Pustelnik and Raciborz records shifting the time series by the calculated Lagrangian travel times. Then, we estimate the shaft-wise emissions by the Philipps-Tikonov technique.





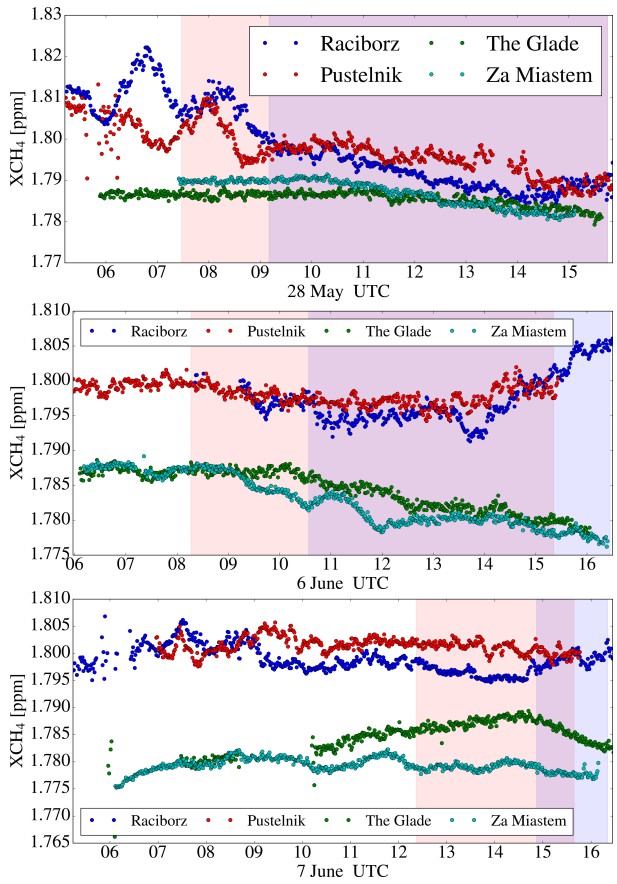

**Figure 7.** XCH$_4$ for 28 May (top), 6 June (center), and 7 June (bottom), 2018, at the stations Raciborz (blue), Pustelnik (red), The Glade (green), and Za Miastem (cyan). Background shadings indicate the time frames used for further analysis considering the individual travel times (light red for Pustelnik, blue for Raciborz, purple for both (Raciborz and Pustelnik).

## 5.1 Southern station Pustelnik

Focusing on the Pustelnik records first, Fig. 8 illustrates the measured and simulated $\Delta$XCH$_4$ enhancements (translated into units of kg m$^{-2}$) and the time periods used for estimating emission rates (blue shading). Generally, FLEXPART simulations using the a priori emissions based on E-PRTR underestimate the enhancements substantially for all cases. The FLEXPART simulations with optimized emission rates fit the measurements well. Residual discrepancies range between $2.35 \times 10^{-8}$ kg m$^{-2}$ and $4.56 \times 10^{-8}$ kg m$^{-2}$ ($2.62 \times 10^{-8}$ kg m$^{-2}$ and $4.72 \times 10^{-8}$ kg m$^{-2}$) in terms of mean bias (root mean square error). The regularization strength for optimizing the emissions was determined via the L-curves depicted in Fig. 9, for which the L-shape is well recognizable. The corner of the L-curve and the corresponding regularization parameter $\lambda$ is identified by visual inspection. Beside excluding data at the start of the daily time series due to the travel time, we also exclude data that we diagnosed to be affected by systematic forward model errors. The periods before 10 and after 13:30 UTC on 28 May (Fig.





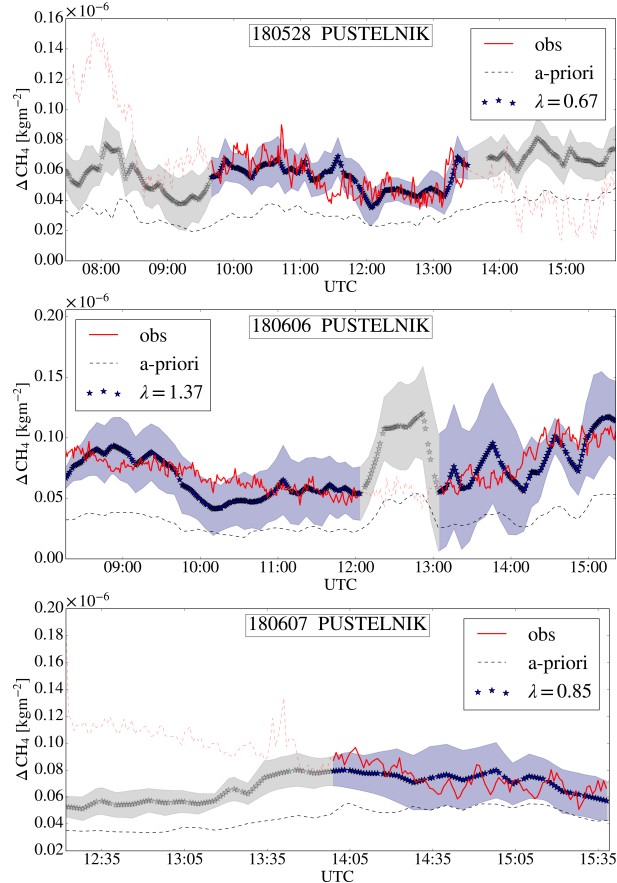

**Figure 8.** $\Delta$XCH$_4$ for 28 May (top), 6 June (center), and 7 June (bottom), 2018, measured (red solid and dotted line) and simulated based on the a priori emissions (grey dashed) and on optimized emissions (dark grey and blue) at the station Pustelnik. The grey and blue shadings indicate the uncertainties due to measurement errors and atmospheric variability.

8, upper panel) are an illustrative example. For these periods, we find substantial deviations between the measured and the modeled timeseries, that the optimization cannot resolve by adjusting emission rates. We expect that such patterns originate from systematic errors in the wind fields that drive FLEXPART. If we include these periods in our optimization scheme, we end up with distorted L-curves (Fig. 9, upper panel, grey dashed line). Thus, inspection of the L-curve provides us with a diagnostic tool to identify periods that are affected by forward model errors which we then exclude. The other excluded periods are around

noon on 6 June and the beginning of 7 June (faint colors in Fig. 8).

After selecting the data periods well represented by the FLEXPART simulations and choosing a suitable regularization parameter, we estimate the compatible shaft-wise emission rates illustrated in Fig. 10. Before evaluating the estimates, it is mandatory to inspect the averaging kernels (upper sub-panels in Fig. 10) which encode the information on whether individual shafts or groups of shafts can be resolved. For 28 May (upper panel in Fig. 10), the averaging kernels are greater than 0.7

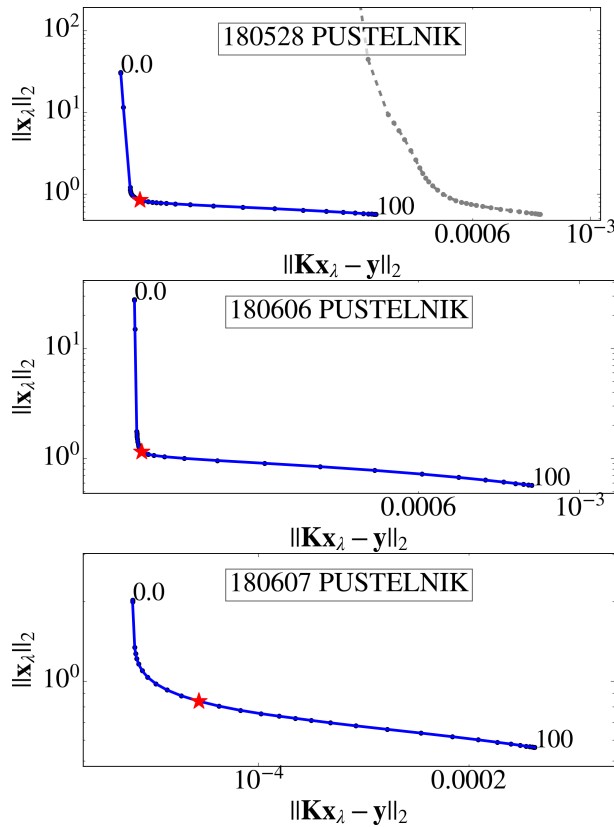

**Figure 9.** The L-curves for the three Pustelnik case studies. The upper panel additionally depicts the L-curve of the same case study (as dashed grey line) but under consideration of the full data set including the morning and afternoon hours which suffered from forward model errors and are omitted in the final analysis (blue curve). The regularization parameters, $\lambda$ range from 0 to 100. The red star marker depicts the respective $\lambda$ used for the emission estimation.

and well positioned for the two Silesia shafts while, for the Brzeszcze shafts, the averaging kernels are small indicating that there is very little measurement sensitivity to these shafts. Low sensitivity might be caused by air mass trajectories going over these shafts only for a short period of time. For the Silesia shafts, the estimated emission rates are substantially greater than the E-PRTR-based prior. On 6 June (middle panel in Fig. 10), the averaging kernels indicate substantial sensitivity to Silesia V, Brzeszcze II, VI, IX and some sensitivity to Brzeszcze IV but the averaging kernels for Brzeszcze II and VI are double-peaked

at each of the shafts indicating the shafts cannot be resolved individually e.g. because trajectories went over both shafts for most of the time. A similar double-peaked averaging kernel occurs for Silesia V and Brzeszcze VI. The shaft that can be best resolved is Brzeszcze IX. Generally, the optimized emissions are again significantly greater than the priors. For Brzeszcze VI, the prior even indicates zero emissions while our estimate exceeds $10\,\text{kt}\,\text{a}^{-1}$. On 7 June (middle panel in Fig. 10), sensitivity is considerable for Silesia I and V and for Brzeszcze IV and, to a lesser degree for Brzeszcze II, but generally the averaging

kernels are not single-peaked but distributed among several shafts which fits the rather unstable wind conditions on that day.

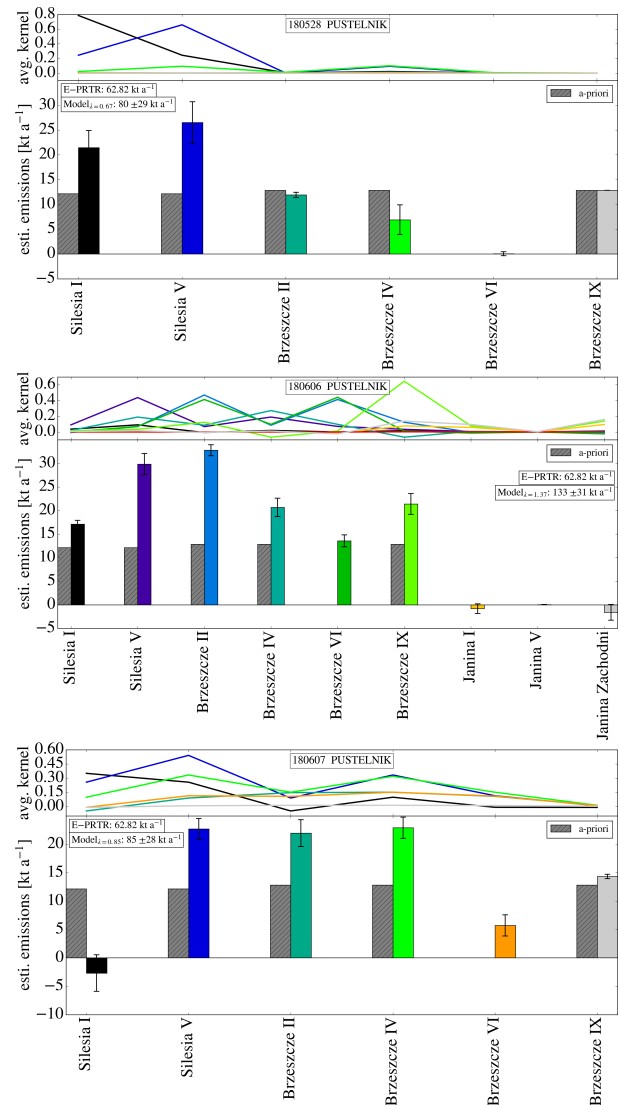

**Figure 10.** Shaft-wise emission estimates (lower sub-panels) for the three case studies at the southern station Pustelnik and the corresponding averaging kernels (upper sub-panels). Colors of the shaft-wise emission estimates resemble colors of the averaging kernel. Error bars contain atmospheric variability and observational uncertainty (measurement noise and background variability). Grey bars illustrate the a priori emission estimates.

It is noteworthy, that we find small emissions for Silesia I on 7 June while for the other days Silesia I showed emissions of $15 - 20 \, kt \, a^{-1}$. This might point at day-to-day variability of the ventilation. Aggregating the emissions over all shafts (Table 2), we find that on all three days the optimized emissions are substantially greater than the prior. On 28 May and 7 June, the total optimized and prior emissions are compatible within the error bars but on 6 June, our estimates indicate emissions twice





as high than the prior, $133 \pm 31 \, kt \, a^{-1}$ compared to $63 \, kt \, a^{-1}$. The difference of the total emission estimates to the other days is largely due to better sensitivity to all the Brzeszcze shafts and to the larger respective emission estimates.

## 5.2    Western station Raciborz

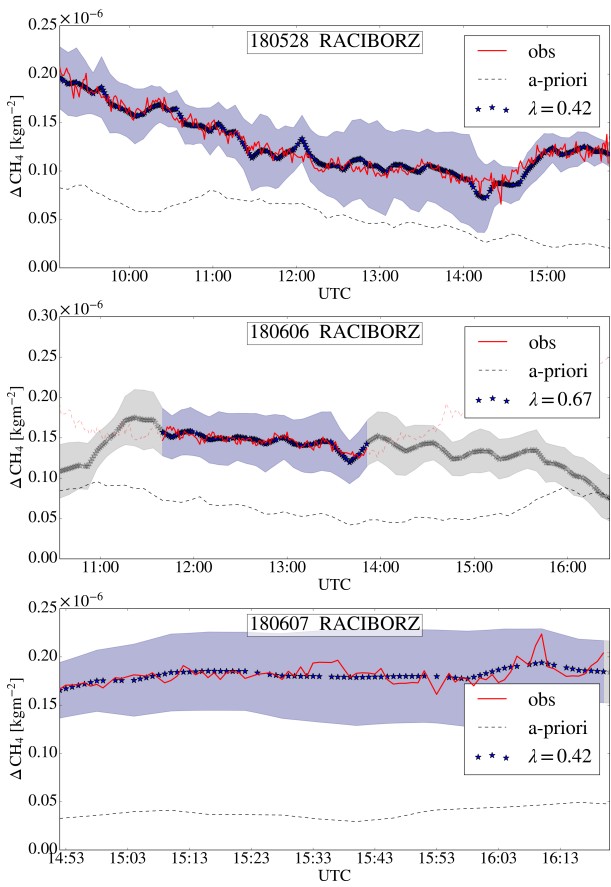

**Figure 11.** $\Delta$XCH$_4$ for 28 May (top), 6 June (center), and 7 June (bottom), 2018, measured (red solid and dashed line) and simulated based on the a priori emissions (grey dashed) and on optimized emissions (dark grey and blue) at the station Raciborz. The grey and blue shadings indicate the uncertainties due to measurement errors and atmospheric variability.

     In contrast to the case studies for Pustelnik, the FLEXPART simulations indicate that the western station Raciborz is influenced by a large and varying number of shafts (between 30 and 50 shafts for the three days discussed here). Figs. 11 and 12

show the fits to the $\Delta$XCH$_4$ observations for the prior and optimized emission estimates and the L-curves for the selection of the regularization parameter, respectively. For 6 June, we again find that the FLEXPART simulations could not represent our measurements at the beginning and end of the time series and therefore, we excluded these periods based on visual inspection of distortions of the L-curve. Overall, after optimization the simulated $\Delta$XCH$_4$ records fit the observations well, while simu-



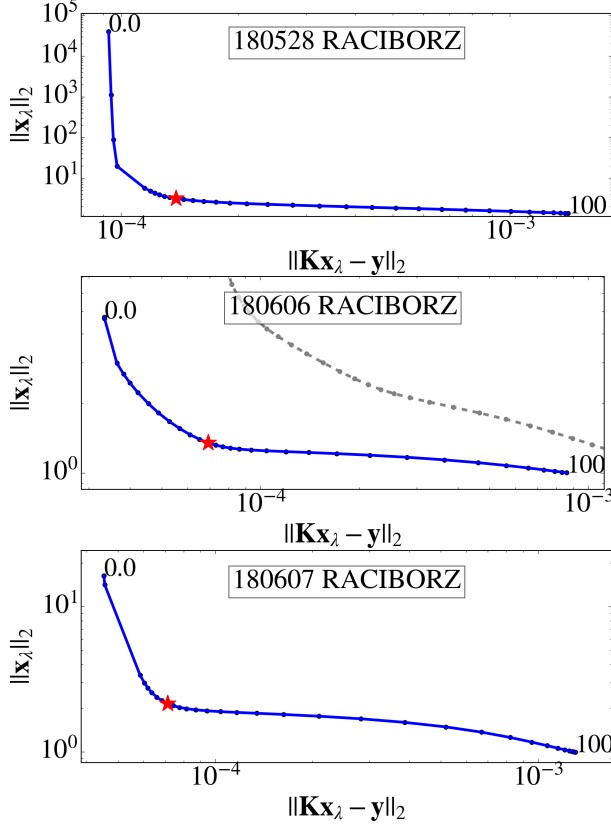

**Figure 12.** The L-curves for the three Raciborz case studies. The middle panel additionally depicts the L-curve of the same case study (as dashed grey line) but under consideration of the full data set including the morning and afternoon hours which suffered from forward model errors and are omitted in the final analysis (blue curve). The regularization parameters, $\lambda$ range from 0 to 100. The red star marker depicts the respective $\lambda$ used for the emission estimation.

lations with prior emissions show substantially smaller-than-observed $\Delta$XCH$_4$. The residual differences between simulations

with optimized emissions and observations range between $7.13 \times 10^{-8}\,\mathrm{kg\,m}^{-2}$ and $1.45 \times 10^{-7}\,\mathrm{kg\,m}^{-2}$ ($7.49 \times 10^{-8}\,\mathrm{kg\,m}^{-2}$ and $1.46 \times 10^{-7}\,\mathrm{kg\,m}^{-2}$) in terms of mean bias (root mean square error).

Fig. 13 illustrates the shaft-wise emission estimates and the corresponding averaging kernels (upper sub-panels). Due to the large number of shafts involved, the picture is less clear than for the Pustelnik station. The averaging kernels show that it is typically groups of shafts to which the measurements have good sensitivity but that, mostly, individual shafts cannot be

resolved. Overall, the optimization points at emissions greater than the prior. When aggregated over all the shafts (Table 2), the optimized emissions are a factor 2.4 (28 May), 1.7 (6 June), 2.4 (7 June) greater than the prior emissions and the differences are greater than the error bars estimated from measurement errors and the ensemble simulations representing transport variability. On 28 May, our estimates indicate large contributions from ventilation associated with the southwestern part of the USCB. On 6 June, the more northern and central mines show emissions greater than the prior which is zero for some of the mines (e.g.





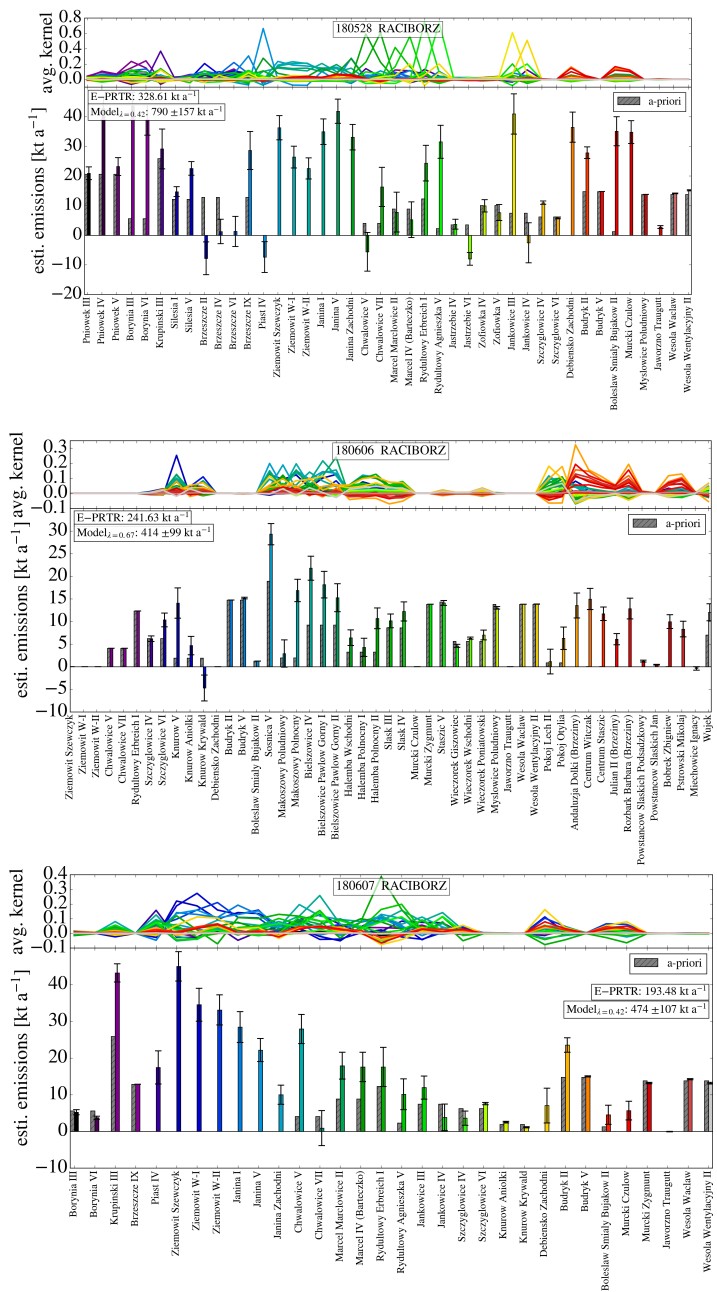

**Figure 13.** Shaft-wise emission estimates (lower sub-panels) for the three case studies at the western station Raciborz and the corresponding averaging kernels (upper sub-panels). Colors of the shaft-wise emission estimates resemble colors of the averaging kernel. Error bars contain atmospheric variability and observational uncertainty (measurement noise and background variability). Grey bars illustrate the a priori emission estimates.





Centrum Witczak/Staszic and Julian II/Rozbark Barbara). On 7 June, the number of contributing shafts is less than for the other days and the estimates point at large emissions from ventilation of Ziemowit, Janina and Chwałowicze V.

| Station | Date | Estimated emissions (kt a$^{-1}$) | Error $\sqrt{\sum \boldsymbol{S}_{x,\lambda}}$ (kt a$^{-1}$) | % | E-PRTR 2014 (kt a$^{-1}$) | Control run residuals (kgm$^{-2} \times 10^{-8}$) BIAS | RMSE | Observational error (kt a$^{-1}$) | % | Background variability (kt a$^{-1}$) | % |
|---|---|---|---|---|---|---|---|---|---|---|---|
| Pus. (South) | 28 May | **80** | **29** | 36 | 62.82 | 2.56 | 2.75 | 0.62 | 0.8 | 0.99 | 1.2 |
| | 6 June | **133** | **31** | 23 | 62.82 | 4.56 | 4.72 | 0.82 | 0.6 | 1.12 | 0.8 |
| | 7 June | **85** | **28** | 33 | 62.82 | 2.35 | 2.62 | 0.65 | 0.8 | 0.83 | 1.0 |
| Rac. (West) | 28 May | **790** | **157** | 20 | 328.61 | 7.13 | 7.49 | 4.46 | 0.6 | 8.5 | 1.1 |
| | 6 June | **414** | **99** | 24 | 241.63 | 8.11 | 8.14 | 3.05 | 0.7 | 2.2 | 0.5 |
| | 7 June | **474** | **107** | 23 | 193.48 | 14.5 | 14.6 | 3.12 | 0.7 | 5.3 | 1.1 |

**Table 2.** Overview of all case studies including respective emission sums and their errors (bold numbers). Mean bias and root mean square error (RMSE) of the `CONTROL` run residuals refer to the regularized solution. The errors due to observational uncertainties and due to background methane variability are listed in the last four columns.

## 6   Discussion & Conclusion

We estimated CH$_4$ emissions from coal mine ventilation facilities in the USCB during the CoMet campaign in May/June 2018. To this end, we deployed four EM27/SUN spectrometers in a network configuration enclosing the USCB. Combining pairs of
upwind/downwind XCH$_4$ observations with trajectory simulations by the Lagrangian particle dispersion model FLEXPART, we attributed the observed $\Delta$XCH$_4$ enhancements to the emission rates of the facilities. The trajectory calculations were driven by wind fields simulated by WRF under assimilation of wind measurements by three wind lidars distributed throughout the region. The trajectory calculations also enabled us to take into account the travel time of airmasses between the upwind and the downwind stations. For emission estimation, we used a Phillips-Tikhonov regularization approach together with the L-
curve criterion for selecting a well-suited regularization parameter. The regularization scheme comes with the averaging kernel diagnostics that allow for quantifying which facilities or groups of facilities the measurements are sensitive to. For estimating errors, we constructed an ensemble of seven FLEXPART runs, each with slightly perturbed atmospheric parameters (wind speed, wind direction, PBL height).

Upscaling our emission estimates to annual emissions, we find higher emission rates than listed by E-PRTR. Other studies
(Luther et al., 2019; Kostinek et al., 2020; Fiehn et al., 2020) have shown better agreement between their instantaneous estimates and the E-PRTR inventory, while, in our study, only two out of six cases are compatible with the E-PRTR inventory within the error range (cf. Table 2). The other four cases suggest 1.7 to 2.4 times higher emissions than reported by the E-PRTR.





The comparisons with E-PRTR are only of illustrative nature. Depending on the under-ground mining activities, the emissions from the mining process are highly variable in time and thus, our measurements are certainly not representative of the annual
total reported by E-PRTR. In order to constrain the latter, a permanent observatory network would need to be operated with a reasonably dense sampling throughout the year.

Our emissions estimates for the totals of all contributing facilities show errors in the range between 23% and 36%. Measurement errors (0.62 to $4.46\,\mathrm{kt\,a^{-1}}$) and backgorund variability induced errors (0.83 to $8.5\,\mathrm{kt\,a^{-1}}$) are negligibly small compared to errors induced by uncertainties of the wind fields (27 to $143\,\mathrm{kt\,a^{-1}}$). As hinted at by the averaging kernels, estimates for
individual shafts are correlated and these correlations were taken into account when calculating the aggregated total emissions. For all cases, these background concentrations were taken from the measurements conducted at the eastern station The Glade since the trajectory calculations showed that it was more representative for the background than the northern station Za Miastem. While for 28 May and 6 June, the two stations show similar concentrations (Fig. 7), The Glade records higher $XCH_4$ than Za Miastem on 7 June. Thus, the background concentration field has some spatial variability which might be connected to
remote sources such as the Krakow urban region affecting The Glade but not Za Miastem (Menoud et al., 2021). In the future, we will aim at estimating errors due to spatial background variability by running a model that includes all known sources in a larger area around the target region or by deploying more spectrometers.

Our inverse method was set up such that the parameters to be estimated included all the individual $CH_4$ ventilation shafts for which the trajectory calculations indicated contributions to the measurements. The problem requires regularization since
$CH_4$ plumes might mask each other (e.g. shafts located behind another shaft along the trajectory), since the detected contributions might have occurred only for a short period, and since, due to errors in the wind fields, the trajectories might indicate contributions when there were actually none or vice versa (e.g. for trajectories barely hitting the downwind station). The Philips-Tikhonov regularization with L-curve criterion provides useful tools to diagnose the robustness and information content of the problem. The L-curve shows distortions for episodes when the FLEXPART trajectories cannot represent the true
link between emissions and downwind concentrations. Thus, inspecting the L-curve allows for excluding episodes affected by these simulation errors. The averaging kernels are useful to identify the shafts or groups of shafts for which the emissions can be reliably estimated from the observations. For our configurations, we found that we can resolve emissions of individual shafts for the Pustelnik downwind station where only a few comparatively close facilities contribute to the $\Delta XCH_4$ enhancements. For the Raciborz station with a much larger catchment area, we typically can resolve groups of shafts along the main wind
direction but disentangling individual shafts is not possible when they are close to each other or behind each other along the airmass trajectory. Nonetheless, we argue it is useful to set up the problem in terms of individual shafts instead of a priori aggregating shaft clusters since, inspecting the averaging kernels, provides a tool to check which shafts can be resolved and since the aggregated total emissions and their errors can be calculated a posteriori.

$CH_4$ emissions other than from coal mining are not considered in our FLEXPART simulations i.e. the pair-wise $\Delta XCH_4$
enhancements are assumed to be caused by only the ventilation shafts. Kostinek et al. (2020) have inspected EDGAR v4.3.2 for $CH_4$ emissions from sources that are not related to coal-mining activities such as landfills, waste water treatment and livestock in the USCB. They found that these other sources amount to roughly 14% of the USCB total $CH_4$ emissions and that the



larger contributions stem form the northwest of the basin which our measurements are not sensitive to. The CoMet inventory (Gałkowski et al., 2021) lists annual emission estimates for four from 16 landfills relevant for our case studies, which in total

amount to $0.97\,\mathrm{kt\,a^{-1}}$. We expect similar emission estimates for the other twelve landfills reported with undefined emissions. Thus, we assume that contributions from sources other than coal mining are small compared to our error bars.

Overall, our study shows that deploying sun-viewing spectrometers in an ad-hoc network configuration around the USCB allows for estimating $CH_4$ emissions from coal mine ventilation facilities with some resolution for individual facilities and groups of them, depending on the deployment location. Given that the errors are dominated by uncertainties in the wind fields

driving the trajectory calculations, it is essential to validate model winds by local wind data or, as in our case, to assimilate local wind lidar measurements. In the view of developing such networks toward a monitoring capacity, priority should be put on making the networks permanent for better temporal representativeness of the observations and on making the networks denser in order to gain sensitivity to more shafts and to better quantify spatial variability in the background concentrations.

## 7   Data availability

The data are available from the author upon request.

*Author contributions.* AL and AB wrote the paper. AL, JK, RK, LS, SW, SD, MS, AF, AD, and DD operated the EM27/SUN spectrometers in the field during the campaign and collected and shared the data. NW provided the wind lidar data. AB, FH, MF, JC, and FD supported preparations for the measurement campaign, contributed to the spectral retrievals, and assisted with

data postprocessing. JN and JS provided detailed information about coal mining and $CH_4$ ventilation and functioned as local advisers. CK and JK provided the WRF and FLEXPART framework. SV contributed to the discussion. AB and AR developed the research question.

*Competing interest.* The authors declare that they have no conflict of interest.


*Acknowledgements.* The Heidelberg team acknowledges support by the Heidelberg Center for the Environment and funding by the Excellence Strategy - a funding programme of the Federal and State Governments of Germany. Further, we acknowledge funding for the CoMet campaign by BMBF (German Federal Ministry of Education and Research) through AIRSPACE (grant no. FKZ: 01LK1701A). We thank DLR VO-R for funding the young investigator research group "Greenhouse Gases". This

work used resources of the Deutsches Klimarechenzentrum (DKRZ) granted by its Scientific Steering Committee (WLA) under project IDs 1104 and 1170.


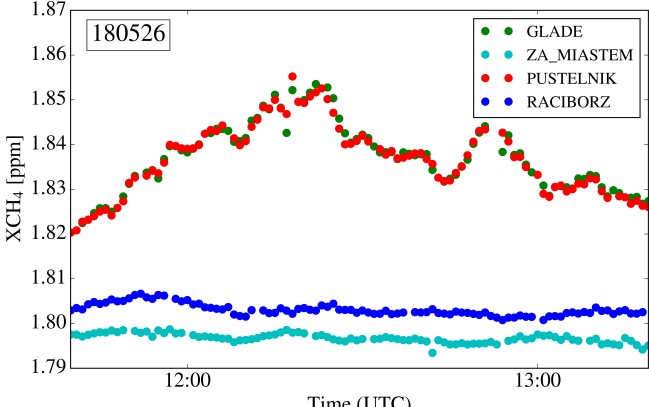

**Figure A1.** Calibrated XCH$_4$ measurements from 26 May. The instruments The Glade and Pustelnik measured side-by-side at the station Pustelnik. The other two instruments measured at their respective campaign locations. The calibration of The Glade towards the other instruments is based on these measurements.

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
