# Peer review of "Observational constraints on methane emissions from Polish coal mines using a ground-based remote sensing network"

_Atmospheric Chemistry and Physics, 2021_

## Author Comment (AC1)

Dear Brad Weir,

Thank you very much for your review! All your points are useful and improve the manuscript a lot. We answered all questions and implemented your suggestions. Answers are written in italics, changes regarding the manuscript are written in blue italics.

**Specific points**

1. A correction after COCCON calibration? I found the correction applied after the COCCON calibration to be confusing/worrying. Isn't one goal of the COCCON calibration to prevent issues like these? Obviously this is more a question for the COCCON group than for this paper, but many of those people are coauthors, so it seems relevant. Could you please clarify here?

*The centralized COCCON calibration activities as reported by Frey et al. (2019), and Alberti et al. (2021) are of utmost importance in order to achieve optimal performance of the EM27/SUN network and for the proper tying of this network as a whole to the TCCON scale. Although the EM27/SUN spectrometer is very robust, we cannot exclude that repeated transports might slightly alter the optical alignment. The EM27/SUN instruments that were operated during the CoMet campaign were used for different measurement campaigns around the globe and therefore transported and operated in a variety of different locations after their previous centralized calibration check by the central facility. Therefore it is advisable to perform a dedicated side-by-side calibration in the framework of a campaign which focuses on the detection of differential column signals using several distributed spectrometers and to use the resulting relative calibration factors for the interpretation of the observations. This is a recommended and common procedure applied during similar network deployments to exclude spurious gradients (Frey et al., 2015; Hase et al., 2015; Chen et al., 2016; Frey et al., 2019; Vogel et al., 2019; Makarova et al., 2020; Dietrich et al., 2021; Jones et al., 2021).*

2. Independent data evaluation. I would've preferred an evaluation against indepedent data, perhaps say a comparison of prior and posterior $CH_4$ against the aircraft data from Kostinek et al. (2021). The set-up in this paper is so close to that paper, it really is puzzling how they disagree on greater vs. lesser emissions for the June 6th case. It would be interesting to see how well they agree with each other in $CH_4$ mixing ratio space and if it has anything to do with the above rescaling. However, the authors can't be expected to do everything, the paper is quite thorough, and the scientific evaluation in Section 4 and the Conclusions, especially the discussion in the context of Kostinek et al. (2021) is sufficient. Since this is a paper about a campaign and network, I thought it might be nice to have a few more sentences on how to resolve these discrepancies other than adding more and more data until they go away.

*Within CoMet we aimed at comparing multiple methane emission estimation approaches. Beside the ground-based teams, various airborne datasets were collected, e.g. flask samples, methane lidar measure-*

[Figure]

**Figure 1.** Comparison of shaft-wise emission estimates reported in this study with estimates by Kostinek et al. (2020) for June 6, 2018. The estimated total sums for the listed shafts are $298\,\mathrm{kt\,a}^{-1}$ for E-PRTR, $295\,\mathrm{kt\,a}^{-1}$ for Kostinek et al. (2020), and $545 \pm 73\,\mathrm{kt\,a}^{-1}$ for this study.

*ments, and imager observations (Krautwurst et al., 2021). Combining all these data will be challenging, which could also be a take-away message from this work. Fig. 1 directly compares the shaft-wise results*
45 *for June 6 from our study to those from Kostinek et al. (2020). Note, that our estimates in Fig. 1 are assembled from both, the southern (Pustelnik) and the western (Raciborz) station, with observational periods from around 8 UTC to 15:30 UTC with one hour gap during noon for the southern station and only roughly 2 hours of data around noon for the western station. The emission estimates by Kostinek et al. (2020) are based on two flights, one in the morning, and one in the afternoon. Since these two data*
50 *sets are sampled on the same day but not during the same time periods, it is possible, that variability of the mining operations contributes to the discrepancies, and it is also clear that our ground-based stations are sensitive to shafts that the aircraft data of Kostinek et al. (2020) are not, and vice versa. Overall, while our study intends to demonstrate how to measure $CH_4$ emissions from mining facilities using an EM27/SUN network, it is evident that only long-term observations covering various observation*
55 *techniques can enable emission monitoring.*

3. EDGAR version and year. The year 2017 on line 32 must be a typo because EDGAR v4.3.2 ends in 2012. I hate to be this reviewer, but that is a very old version of EDGAR. The current version, v6.0, actually includes the study year of 2018. It also includes a seasonal cycle, which could be relevant to the
60 results in this paper. My understanding is that E-PRTR and not EDGAR is used in the study's prior. In that case, it should be straightforward to include the number cited in the text for the newest version and the study year. If EDGAR was used in the prior, I'd be happy if you just noted both numbers (2012 from

v4.3.2 and 2018 from v6.0), no need to redo experiments. Also, it might be nice to list the exact sectoral breakdown that was used.

65

*Indeed, the EDGAR paragraph contains errors and is rewritten. We apologize for this negligence. Since EDGAR is not used as prior (but E-PRTR plus updates), this does not affect the emission estimates. The respective paragraph is rewritten: EDGAR v4.3.2 (Emission Database for Global Atmospheric Research) accounts emissions of $675\,kt\ CH_4\ a^{-1}$ (Janssens-Maenhout et al., 2017) for fuel exploitation emissions*

70 *(EDGAR abbreviation PRO) for the USCB in the year 2012. EDGAR v6.0 (Crippa et al., 2020) states $454\,kt\ CH_4\ a^{-1}$ for the sector IPCC 1B1a (fuel exploitation COAL) for the USCB in 2018.*

4. Satellites. Campaigns like CoMet and networks like COCCON have another very obvious use in that they are essential in the bias correction and interpretation of retrievals from satellites. There are

75 now several satellites in orbit that observe $CH_4$, e.g., the GOSATs, TROPOMI, with the expectation of several more (GeoCarb, Merlin) including several from commercial partners. Could you dedicate at least a sentence or two, perhaps in addressing point 2 above, to describe what impact your results might have on those missions and vice-versa?

80 *It was part of our network design to eventually compare our observations to TROPOMI and/or GOSAT data. However, the satellite data (after cloud and quality screening) were too sparse to allow for a profound comparison. A dedicated satellite validation campaign with a long term stationary network deployment could indeed contribute to improvements on both sides, the satellite and the ground-based emission estimation. We added the following sentence to the discussion: In addition to the monitoring*

85 *aspect, permanent networks could also validate satellite missions and the different ground-based and space-borne emission estimation approaches could be consolidated.*

5. The background variability of 0.6 ppb on line 178 seems an order of magnitude too small. Could you provide more support for why this is reasonable? Looking at Table 3 from the Barkley et al. (2017) paper

90 from ACT-America, they have day-to-day variability in background $CH_4$ over North America of several ppb. I'm not sure that exactly maps to the purposes here, but assuming much of that is passing weather patterns, I'd guess a number of at least 6 ppb would be more appropriate. Maybe this gets picked up in the L-curve fitting, so it isn't of much practical importance? Either way, 0.6 ppb seems low.

95 *Our wording was confusing in the respective paragraph. We assume $0.6\,ppb$ for the measurement noise based on our spectrometric observation. The "background variability" (we should have called it "background error") is derived from a sensitivity study for each individual case by shifting the airmass travel time between upwind and downwind location by plus-minus 30 min. The resulting background error ranges between $0.3\,ppb$ and $2.2\,ppb$. Since we calculate the background and its error directly from*

100 *the daily station data, day-by-day methane background variability does not play a role. For clarification, we rewrote the paragraph: The estimated background error ranges between 0.3 ppb and 2.2 ppb for the individual case studies, based on the differences of minimum and maximum average $\Delta XCH_4$ calculated under consideration of $\pm 30\,min$ time shifts of the simulated methane travel times. The measurement noise amounts to 0.6 ppb, calculated as the standard deviation of the averaged measurements of The Glade (E)*

 *and Za Miastem (N) from 7 UTC to 10 UTC on 28 May 2018 (see top panel in Fig. 7.)*

**Technical points**

1. Can you give some indication of the correlation lengths/patterns of model background ($\mathbf{S}_\epsilon$)? Correlation lengths affect the interpretation of the values on the diagonal, so this would be helpful putting those numbers in the context of other studies. Since the authors used an ensemble approach, a correlation "length" might not be 100% appropriate, but I'm sure they can come up with some number/figure to indicate how errors decorrelate horizontally, even if it is just one number or a range.

*Our observation error covariance matrix $\mathbf{S}_\epsilon$ is diagonal representing the measurement error from the $XCH_4$ retrieval, the transport model error from our sensitivity studies and the background error. The manuscript states this in section 4 after equation (1). So, we did not consider observational error correlations. Following the reviewer comment, we looked into an easy way to estimate observation error correlations and combine them with the L-curve technique for regularization. We found no straightforward way how to come up with reliable error correlations without substantially investing in ensemble calculations for every single case. Also, the L-curve appeared distorted for the attempts undertaken. Therefore, we propose to postpone the assessment to future studies. We added the following statement to section 4: Note that, for simplicity, we did not consider correlations in $\mathbf{S}_\epsilon$.*

2. At least two words about chemistry. Given the short time scale of the experiments in this paper, oxidation probably doesn't play much/any of a role, but it might be worthwhile to say that somewhere?
*We added the following sentence to the discussion: We did not consider chemical reactions as e.g. oxidation of methane by $OH$, since the time scales relevant for this study are too short that oxidation would have a significant influence on the emission estimations.*

3. "An" averaging kernel instead of "the". I found the use of the averaging kernel very helpful, but it might be better to call it "an" averaging kernel instead of "the" so that the reader does not confuse it with the averaging kernel of the EM27/SUN retrieval. Either that or qualifying adjectives like "the emissions averaging kernel" might help.
*We changed "the averaging kernel" to "the emissions averaging kernel".*

4. For those of us that haven't dedicated a good chunk of our life to this campaign over the past few years, it might be helpful to add the cardinal directions to the site names in the text as is done in Table 1. I did find myself going back and forth a lot trying to figure out which station was where, but got the hang of it once I finished.
*This is a very helpful advise! We added the cardinal directions.*

**References**

Alberti, C., Hase, F., Frey, M., Dubravica, D., Blumenstock, T., Dehn, A., Surawicz, G., Harig, R., Orphal, J., and the EM27/SUN-partners team: Improved calibration procedures for the EM27/SUN spectrometers of the COllaborative Carbon Column Observing Network (COC-CON), Atmospheric Measurement Techniques Discussions, 2021, 1–48, https://doi.org/10.5194/amt-2021-395, 2021.

[revised manuscript text omitted]

---

## Author Comment (AC2)

Dear referee,

Thank you very much for your review! Answers are written in italics, changes regarding the manuscript are written in blue italics.

**Reviewer comments:**

Line 45: "The setup largely mimics previous network deployments for quantifying urban greenhouse gas emissions in Berlin (Hase et al., 2015), Paris (Vogel et al., 2019), St. Petersburg (Makarova et al., 2020), Munich (Dietrich et al., 2021), Indianapolis (Jones et al., 2021) and other places."

This statement makes me ask:

What were the major findings from these deployments?

*The mentioned studies generally show the technical feasibility of EM27/SUN networks and they report on emission estimates with a focus on urban settings. Generally, there is a technical evolution for the instrumental part from the early demonstrator studies such as in Hase et al. (2015) towards the latest installation of a long-term almost autonomous urban network in Dietrich et al. (2021). There is also an evolution of the analytical techniques used for inferring emissions. Essentially, it is still an open question - also somewhat depending on the location - how to best set up meteorological models and inverse techniques to deliver reliable emission estimates. All locations come with their own configurations in terms of domain size, topography, meteorological situation and thus, they require a dedicated setup of the instruments and of the data interpretation tools. The quoted sentence serves the purpose to list the previous work that our study builds on. Our study is unique in that it addresses the USCB, the largest source of fossil $CH_4$ in Europe characterized by dozens of strong point sources for which we deliver "instantaneous" emission estimates. Our study is also unique in that it uses a trajectory modelling and inverse estimation approach that has not been used before in the context of these ground-based networks. For clarification we added the following sentence to the introduction:* This work demonstrates emission estimation of a methane emitting hotspot based on a FTS network combined with a Phillips-Tikhonov regularized inversion approach.

As shown in the previous deployments, what are the strengths and limitations of your methodology for constraining emissions?

*Generally, a particular strength of the EM27/SUN networks is that they deliver precise and accurate total columns i.e. unlike for in-situ ground-based or aircraft measurements, there is no need to account for the "unobserved" portion of $CH_4$ (or $CO_2$) plumes above or below the sampling point. Another strength is that, if the network nodes are arranged along the prevailing wind direction, we can directly*

*pair upwind-downwind measurements and infer the enhancements due to the local emissions in-between the nodes. A weakness is that the EM27/SUNs require sunlight and fair weather conditions which reduces the sampling density. Some earlier studies also suffered from poor knowledge of the wind fields which is typically the largest error source for the emission estimates. In that context, there is a need to further improve on the data interpretation in terms of using meteorological models and inverse techniques to estimate emissions from the pair-wise concentration gradients.*

Have you addressed the limitations that were identified in previous deployments? That is, is your setup better than earlier setups?

*The focus of our study is not only methodological. We want to report on a successful independent estimation of $CH_4$ emissions from Europe's fossil $CH_4$ hotspot and on the methods how we have achieved this. The key technical novelties of our study are the following: We deployed 3 windlidars together with the EM27/SUNs i.e. we have excellent knowledge on the wind fields. We employed state-of-the-art meteorological modelling (WRF, wind lidar data assimilation, FLEXPART) and we used an inverse estimation technique that allows for rigorously discussing the information content of our data. Combining these aspects, we believe that, also from a methodological perspective, our study is a useful and original addition to previous work.*

Is there anything new or novel about your setup relative to the previous ones?

*See previous comment.*

Is the importance of this manuscript mainly in the application of the setup to a new emissions source?

*Following our comments above, we see two key messages: 1. We deliver emission estimates for $CH_4$ from coal mining in the USCB. These are estimates of instantaneous emissions and thus, comparison to the annual E-PRTR reporting is questionable, but our estimates tend to be larger than the E-PRTR analogues and they are also different from previous campaign reports (Luther et al., 2019; Kostinek et al., 2020). That is, if we want to be certain of how much $CH_4$ comes out of the USCB, we need longterm measurements by various techniques. 2. The methods that we have used are rigorous with respect to the quality of wind information (3 wind lidars assimilated into WRF) and with respect to the analysis of the information content (FLEXPART trajectories, Tikhonov inverse method with averaging kernels). So, we suggest that others consider using similar approaches.*

Last paragraph of Section 6: Discussion and Conclusions

Since this setup has been used before, how feasible (e.g., from a cost perspective) would it be to use this setup to quantify major emission sources around the world? That is, could it be easily commercialized? Or is this setup primarily for scientific research? My motivation for these questions is the need for affordable options for constraining $CO_2$ and methane emissions given that satellites, especially for observing $CO_2$,

have limitations that do not always allow for reliable space-based constraints.

85

*The MUCCNET (Dietrich et al., 2021) is probably quite close to a commercialized setup. It requires at least 4 EM27/SUNs to cover the cardinal directions with respect to the target location. The EM27/SUNs typically do not require frequent maintenance and the instrument operations are quasi-autonomous such that the network can be handled by a single person. Moreover, the COCCON data processing tools are*

90 *freely available to any potential users. Both the operation of the spectrometers and the subsequent data analysis are much simpler than the operation of a TCCON site, which definitely requires engineering and scientific expertise. Thus, we believe that EM27/SUN networks can be useful and cost-effective addition to monitoring systems for greenhouse gas emissions operated by , e.g., environmental agencies. The cost of an EM27/SUN spectrometer is in the order of magnitude of TDLAS based in-situ sensors applied for*

95 *high precision measurements of atmospheric greenhouse gas abundances.*

In the vein of satellites, have you compared your results to space-based constraints, especially for methane? Could your method be used to validate satellite emission constraints?

100 *For the study in the USCB put forward here, we found that the satellite records from TROPOMI und GOSAT were too sparse (after cloud and quality screening) to allow for a meaningful comparison. But, we have conducted satellite validation studies with our ship-borne variant of the EM27/SUN (Klappenbach et al., 2015; Knapp et al., 2021; Butz et al., 2022). In particular, Butz et al. (2022) is a review for use cases of mobile EM27/SUNs including satellite validation aspects.*

**105 References**

Butz, A., Hanft, V., Kleinschek, R., Frey, M. M., Müller, A., Knapp, M., Morino, I., Agusti-Panareda, A., Hase, F., Landgraf, J., Vardag, S., and Tanimoto, H.: Versatile and Targeted Validation of Space-Borne $XCO_2$, $XCH_4$ and XCO Observations by Mobile Ground-Based Direct-Sun Spectrometers, Front. Remote Sens., 2, 775 805, https://doi.org/10.3389/frsen.2021.775805, 2022.

Dietrich, F., Chen, J., Voggenreiter, B., Aigner, P., Nachtigall, N., and Reger, B.: MUCCnet: Munich Urban Carbon Column network, Atmospheric Measurement Techniques, 14, 1111–1126, https://doi.org/10.5194/amt-14-1111-2021, 2021.

Hase, F., Frey, M., Blumenstock, T., Groß, J., Kiel, M., Kohlhepp, R., Mengistu Tsidu, G., Schäfer, K., Sha, M. K., and Orphal, J.: Application of portable FTIR spectrometers for detecting greenhouse gas emissions of the major city Berlin, Atmos. Meas. Tech., 8, 3059–3068, https://doi.org/10.5194/amt-8-3059-2015, 2015.

Jones, T. S., Franklin, J. E., Chen, J., Dietrich, F., Hajny, K. D., Paetzold, J. C., Wenzel, A., Gately, C., Gottlieb, E., Parker, H., Dubey, M., Hase, F., Shepson, P. B., Mielke, L. H., and Wofsy, S. C.: Assessing urban methane emissions using column-observing portable Fourier transform infrared (FTIR) spectrometers and a novel Bayesian inversion framework, Atmospheric Chemistry and Physics, 21, 13 131–13 147, https://doi.org/10.5194/acp-21-13131-2021, 2021.

Klappenbach, F., Bertleff, M., Kostinek, J., Hase, F., Blumenstock, T., Agusti-Panareda, A., Razinger, M., and Butz, A.: Accurate mobile remote sensing of $XCO_2$ and $XCH_4$ latitudinal transects from aboard a research vessel, Atmos. Meas. Tech., 8, 5023–5038, https://doi.org/10.5194/amt-8-5023-2015, 2015.

Knapp, M., Kleinschek, R., Hase, F., Agustí-Panareda, A., Inness, A., Barré, J., Landgraf, J., Borsdorff, T., Kinne, S., and Butz, A.: Shipborne measurements of $XCO_2$, $XCH_4$, and XCO above the Pacific Ocean and comparison to CAMS atmospheric analyses and S5P/TROPOMI, Earth System Science Data, 13, 199–211, https://doi.org/10.5194/essd-13-199-2021, 2021.

Kostinek, J., Roiger, A., Eckl, M., Fiehn, A., Luther, A., Wildmann, N., Klausner, T., Fix, A., Knote, C., Stohl, A., and Butz, A.: Estimating Upper Silesian coal mine methane emissions from airborne in situ observations and dispersion modeling, Atmospheric Chemistry and Physics Discussions, 2020, 1–24, https://doi.org/10.5194/acp-2020-962, 2020.

Luther, A., Kleinschek, R., Scheidweiler, L., Defratyka, S., Stanisavljevic, M., Forstmaier, A., Dandocsi, A., Wolff, S., Dubravica, D., Wildmann, N., Kostinek, J., Jöckel, P., Nickl, A.-L., Klausner, T., Hase, F., Frey, M., Chen, J., Dietrich, F., Nęcki, J., Swolkień, J., Fix, A., Roiger, A., and Butz, A.: Quantifying $CH_4$ emissions from hard coal mines using mobile sun-viewing Fourier transform spectrometry, Atmospheric Measurement Techniques, 12, 5217–5230, https://doi.org/10.5194/amt-12-5217-2019, 2019.

Makarova, M. V., Alberti, C., Ionov, D. V., Hase, F., Foka, S. C., Blumenstock, T., Warneke, T., Virolainen, Y., Kostsov, V., Frey, M., Poberovskii, A. V., Timofeyev, Y. M., Paramonova, N., Volkova, K. A., Zaitsev, N. A., Biryukov, E. Y., Osipov, S. I., Makarov, B. K., Polyakov, A. V., Ivakhov, V. M., Imhasin, H. K., and Mikhailov, E. F.: Emission Monitoring Mobile Experiment (EMME): an overview and first results of the St. Petersburg megacity campaign-2019, Atmospheric Measurement Techniques Discussions, 2020, 1–45, https://doi.org/10.5194/amt-2020-87, 2020.

Vogel, F. R., Frey, M., Staufer, J., Hase, F., Broquet, G., Xueref-Remy, I., Chevallier, F., Ciais, P., Sha, M. K., Chelin, P., Jeseck, P., Janssen, C., Té, Y., Groß, J., Blumenstock, T., Tu, Q., and Orphal, J.: $XCO_2$ in an emission hot-spot region: the COCCON Paris campaign 2015, Atmospheric Chemistry and Physics, 19, 3271–3285, https://doi.org/10.5194/acp-19-3271-2019, 2019.